

# Oncolytic effects of the recombinant Newcastle disease virus, rAF-IL12, against colon cancer cells in vitro and in tumor-challenged NCr-Foxn1nu nude mice

Syed Umar Faruq Syed Najmuddin[1,2], Zahiah Mohamed Amin[1,2], Sheau Wei Tan[1,2], Swee Keong Yeap[3], Jeevanathan Kalyanasundram[1], Abhimanyu Veerakumarasivam[4], Soon Choy Chan[5], Suet Lin Chia[1,2], Khatijah Yusoff[1,6] and Noorjahan Banu Alitheen[1,2]

[1] Universiti Putra Malaysia, Serdang, Malaysia
[2] Institute of Bioscience, Universiti Putra Malaysia, Serdang, Malaysia
[3] Xiamen University, Sepang, Malaysia
[4] Sunway University, Subang Jaya, Malaysia
[5] Perdana University, Serdang, Malaysia
[6] Malaysian Genome Institute, National Institute of Biotechnology Malaysia, Kajang, Malaysia

Corresponding author
Noorjahan Banu Alitheen,
noorjahan@upm.edu.my

## ABSTRACT

Colon cancer remains one of the main cancers causing death in men and women worldwide as certain colon cancer subtypes are resistant to conventional treatments and the development of new cancer therapies remains elusive. Alternative modalities such as the use of viral-based therapeutic cancer vaccine is still limited, with only the herpes simplex virus (HSV) expressing granulocyte-macrophage colony-stimulating factor (GM-CSF) or talimogene laherparepvec (T-Vec) being approved in the USA and Europe so far. Therefore, it is imperative to continue the search for a new treatment modality. This current study evaluates a combinatorial therapy between the oncolytic Newcastle disease virus (NDV) and interleukin-12 (IL-12) cytokine as a potential therapeutic vaccine to the current anti-cancer drugs. Several in vitro analyses such as MTT assay, Annexin V/FITC flow cytometry, and cell cycle assay were performed to evaluate the cytotoxicity effect of recombinant NDV, rAF-IL12. Meanwhile, serum cytokine, serum biochemical, histopathology of organs and TUNEL assay were carried out to assess the anti-tumoral effects of rAF-IL12 in HT29 tumor-challenged nude mice. The apoptosis mechanism underlying the effect of rAF-IL12 treatment was also investigated using NanoString Gene expression analysis. The recombinant NDV, rAF-IL12 replicated in HT29 colon cancer cells as did its parental virus, AF2240-i. The rAF-IL12 treatment had slightly better cytotoxicity effects towards HT29 cancer cells when compared to the AF2240-i as revealed by the MTT, Annexin V FITC and cell cycle assay. Meanwhile, the 28-day treatment with rAF-IL12 had significantly ($p < 0.05$) perturbed the growth and progression of HT29 tumor in NCr-Foxn1nu nude mice when compared to the untreated and parental wild-type NDV strain AF2240-i. The rAF-IL12 also modulated the immune system in nude mice by significantly ($p < 0.05$) increased the level of IL-2, IL-12, and IFN-γ cytokines. Treatment with rAF-IL12 had also significantly ($p < 0.05$) increased the expression level of apoptosis-related genes such as Fas, caspase-8, BID, BAX, Smad3 and granzyme B in vitro and in vivo.

Besides, rAF-IL12 intra-tumoral delivery was considered safe and was not hazardous to the host as evidenced in pathophysiology of the normal tissues and organs of the mice as well as from the serum biochemistry profile of liver and kidney. Therefore, this study proves that rAF-IL12 had better cytotoxicity effects than its parental AF2240-i and could potentially be an ideal treatment for colon cancer in the near future.

## INTRODUCTION

A virus is a nanoscale particle ($<10^{-6}$ mm) that can infect the cells of a biological organism such as bacteria, mammals, or plants (*Koudelka et al., 2015*). It carries specific tools on its surface designed to cross the barriers of host cells before consequently delivering their nucleic acid cargo and hijacking the intracellular machinery to produce the components of progeny viruses (*Koudelka et al., 2015*). Engineered viruses may therefore be used in gene therapies aimed at influencing gene expression in living organisms through delivery of integrating or non-integrating exogenous DNA or RNA to treat certain diseases (*Riley & Vermerris, 2017*).

At the turn of the 19th century, oncolytic viruses were identified as possible tumoricidal agents whereby body fluids containing human or animal viruses were used to infect cancer cells (*Kelly & Russell, 2007*). Since then, various oncolytic viruses (OVs) such as the herpes simplex virus (HSV), adenovirus, reovirus, and the Newcastle disease virus (NDV) have been extensively studied for their potential use as anti-cancer agents (*Alemany, 2014*; *Gong et al., 2016*; *Lam et al., 2011*; *Yin et al., 2017*). Non-human viruses like the NDV are sought after as they could retain their oncolytic ability in a host not traditionally susceptible to that particular virus (i.e., viral adaptation for targeting) and at the same time considered to be non-infectious or non-pathogenic in humans. Successful use of the NDV has been demonstrated in various studies using strains such as the MTH68/H, LaSota, PV701 and AF2240-i (*Kelly & Russell, 2007*; *Lam et al., 2011*). The NDV is an avian paramyxovirus that has a 15 kb single-stranded, negative-sense and non-segmented RNA genome comprising six genes that encode for six structural proteins namely, the nucleocapsid protein (NP), phosphoprotein (P), fusion protein (F), envelope matrix protein (M), large protein (L), and hemagglutinin-neuraminidase surface glycoprotein (HN) (*Schirrmacher, 2017*). NDV strains can be classified into three main pathotypes namely, the velogenic (highly virulent), mesogenic (intermediate), and lentogenic (non-virulent) strains (*Zamarin & Palese, 2012*).

The Malaysian viscerotropic-velogenic NDV strain, AF2240-i, was isolated by the Malaysian Veterinary Research Institute (VRI) from a local field outbreak in the 1960s. Due to its high virulence, it has been used as standard challenge virus for the development of local chicken vaccine (*Kalyanasundram et al., 2018*). Interestingly, it has also been reported that this strain exhibits both in vitro and in vivo oncolytic activity against breast,

brain, and cervical cancer cells as well as the non-adherent leukemic cells (*Murulitharan et al., 2013*). With regards to all the oncolytic abilities possessed by the wild-type NDV, genetic engineering has paved its way to maximise the therapeutic efficacy of NDV by arming them with immune-enhancing cytokines. For instance, NDV Anhinga strain expressing interleukin-2 (IL-2) could effectively inhibit the growth of hepatocellular carcinoma in vivo while rNDV-IL2-TRAIL could significantly enhance the induction of apoptosis in cancer cells (*Wu et al., 2016*; *Bai et al., 2014*). Other researchers have also utilized the NDV to express interleukine-7 (IL-7) and interleukine-15 (IL-15) in their studies where the recombinant NDV strain LX/IL7/IL15 showed antitumor activity against murine melanoma cells (*Xu et al., 2018*).

On a related note, the use of recombinant NDV is advantageous over the wild type strain as several studies have demonstrated that IL-12-expressing OVs improve the therapeutic index in pre-clinical tumor models and improve tumor clearance (*Alkayyal, Mahmoud & Auer, 2016*). IL-12 has long been touted as an ideal candidate for tumor immunotherapy due to its ability to activate both innate and adaptive immune system cells during antigen presentation (i.e., provide the bridge/interconnection between innate and adaptive immune system) by aiding the activation and regulation of several immune cells such as macrophages and natural killer cells (*Lasek, Zagożdżon & Jakobisiak, 2014*; *Tsai et al., 2016*; *Tugues et al., 2015*). IL-12 is a heterodimeric cytokine containing a 35 kDa and a 40 kDa subunit that is produced mainly by phagocytic cells such as dendritic cells and macrophages (also known as antigen-presenting cells, APCs) in response to antigenic stimulation. (*Lee & Margolin, 2011*; *Tugues et al., 2015*). Furthermore, IL-12 has previously been used as an immunomodulatory agent in recombinant virus such as adenovirus and efficiently manifested the anti-tumoral effect towards prostate cancer (*Alkayyal, Mahmoud & Auer, 2016*).

In this study, we test the efficacy of recombinant AF2240-i expressing human interleukin-12 (rAF-IL12) in combating colon cancer cells by evaluating both in vitro and in vivo cytotoxicity effects of the rAF-IL12 against HT29 colon cancer cells, compared to the parental wild-type, AF2240-i. The utilisation of IL-12 in this study was due to the fact that IL-12 is a successful anti-tumor agent in previous preclinical studies (i.e., applied in dozens of experimental models in mice involving solid tumors and hematologic malignancies) (*Lasek, Zagożdżon & Jakobisiak, 2014*). IL-12 is also associated with the production and secretion of another potent anti-tumor cytokines namely, interferon-γ and interleukine-2 (*Kalyanasundram et al., 2018*). It is worth to mention that the rAF-IL12 virus is a genetically modified construct of the wild-type NDV strain AF2240-i through the insertion of IL-12 at the M and F intergenic junction, whereas the AF2240-i virus did not possess the IL-12 gene. In the previous study, the rAF-IL12 had been proved to be stable as its concentration did not deteriorate when propagated in the embryonated chicken eggs from passage 1 (starting HA unit of $2^8$) until passage 10 (final HA unit of $2^{11}$). It showed a velogenic virulence nature as it had an intracerebral pathogenicity index (ICPI) of 1.78 (*Amin et al., 2019*). Other than that, the rAF-IL12 also showed cytotoxicity against breast cancer cell lines; MDA-MB-231 and MCF-7 (*Amin et al., 2019*).

## MATERIALS AND METHODS

### Preparation of cell culture

NIH/3T3 normal fibroblast and human HT29 colon cancer cell lines were purchased from the American Type Culture Collection (ATCC) USA. The 3T3 and HT29 cells were cultured in Dulbecco's Modified Eagle's Medium (DMEM) and Roswell Park Memorial Institute (RPMI) 1640 media, respectively. The media were supplemented with 10% fetal bovine serum (FBS) (Gibco, USA) and 1% penicillin/streptomycin (Gibco, USA). The cell cultures were incubated in a humidified incubator at 37 °C in the presence of 5% $CO_2$ and were passaged/subcultured upon reaching 70% confluency as recommended by *Yang et al. (2016)*.

### Preparation of virus

The parental Malaysian viscerotropic NDV strain, AF2240-i, was used as negative control throughout this study. The rAF-IL12 virus was developed in the Virology Laboratory, Faculty of Biotechnology and Biomolecular Sciences, Universiti Putra Malaysia (*Amin et al., 2019*). The viruses were propagated in allantoic fluid of 9–11 day old SPF embryonated chicken eggs and were incubated at 37 °C for 48–72 h. The allantoic fluid was harvested and the titre of the virus was determined by the haemagglutinin assay (HA) using 1% chicken red blood cells. All in vitro assays were performed at 72 h post infection (h.p.i.) (*Amin et al., 2019*).

### 3-[4,5-dimethylthiazol-2-yl]-2,5 diphenyltetrazolium bromide (MTT) assay

The half maximal inhibitory concentration ($EC_{50}$) of the virus towards cancer cells viability was assessed by the MTT assay based on the reduction of yellow tetrazole to purple formazan crystals. Briefly, a concentration of 80,000 cells/ mL was seeded into a 96-well plate and incubated overnight before being infected with serially diluted virus on the following day. Upon reaching the 72 h post infection (h.p.i.), MTT solutions were added before the formazan crystals were solubilized by DMSO. Finally, the absorbance at 570 nm wavelength of the microplate was analysed using the μQuant™ enzyme-linked immunosorbent assay (ELISA) microplate reader (Bio-tek Instruments, Winooski, VT, USA) (*Syed Najmuddin et al., 2016*).

### qPCR validation of virus copy number

RNA was extracted from the rAF-IL12-treated HT29 cells (72 h.p.i.) using TRI Reagent® (Sigma, Saint Louis, MO, USA). The yield and purity of RNA were assessed using Nanodrop machine (Eppendorf, Enfield, CT, USA) before qPCR validation of virus copy number was performed. A Taqman real-time PCR protocol was carried out using the primers 5′-TCCGCAAGATCCAAGGGTCT-3′ and 5′-CGCTGTTGCAACCCCAAG-3′ as well as the TaqMan probe 5′(FAM) AAGCGTTTCTGTCT CCTTCCTCCA (BHQ-3′, which target the fusion (F) gene of the NDV (*Abdolmaleki et al., 2018*). All qPCR reactions were performed in a final reaction volume of 20 μL containing 1x iTaq universal probes reaction mix (Bio-Rad, Hercules, CA, USA), 0.5 μM of each primer, 0.25 μM of

TaqMan probe, 1U of iScript reverse transcriptase, and 300 ng of RNA. A negative control (i.e., without template RNA) was included in each run (*Abdolmaleki et al., 2018*).

## Tissue culture infectious dose (TCID$_{50}$) assay

A concentration of 8,000 cells/ well was seeded into a 96-well plate and incubated overnight before being infected with 10-fold serially diluted viruses on the following day. The infected cells were incubated for five days in a humidified incubator at 37 °C in the presence of 5% $CO_2$. Later, the 96-well plate was stained with 1% crystal violet and the plate was observed under inverted microscope to spot the cytopathic effects caused by the virus. Data was calculated using the Reed and Muench method to determine the TCID$_{50}$/mL (*Jiang et al., 2018*).

## Annexin V FITC analysis

The assay was carried out using Annexin V FITC Kit (BD Pharmigen, San Diego, CA, USA) according to manufacturer's protocol to evaluate the apoptosis induction of rAF-IL12 towards cancer cells. The pellets of rAF-IL12-treated HT29 cells (72-h post-infection) were resuspended in 1x Binding Buffer prior to staining with FITC Annexin V and propidium iodide (PI) dyes. The cell suspension was then allowed to stand in the dark at room temperature for 15 mins before being analysed using the NovoCyte Flow Cytometer (ACEA Biosciences Inc., San Diego, CA, USA) and the NovoExpress® version 1.2.4 software (ACEA Biosciences Inc., San Diego, CA, USA) (*Pawłowska et al., 2018*).

## Cell cycle analysis

The rAF-IL12-treated HT29 cells (72-h post-infection) was subjected to cell cycle analysis using the Cycletest™ Plus DNA Reagent Kit (BD Biosciences, Franklin Lakes, NJ, USA). The cells pellet was first resuspended in solution A (trypsin), followed by in solution B (RNAse A) and finally stained with PI dyes, which requires a 10 mins of incubation period in each step. Later, the mixture was analysed by NovoCyte Flow Cytometer (ACEA Biosciences Inc., San Diego, CA, USA) using NovoExpress® version 1.2.4 software (ACEA Biosciences Inc., San Diego, CA, USA) (*Syed Najmuddin et al., 2016*).

## In vivo animal study

Approximately 5 to 6-week-old male NCr-Foxn1nu nude mice, weighing approximately 20 g were purchased from InVivos Pte Ltd, Singapore. All mice were housed in sterile micro-isolator (filter bonneted and ventilated) cages in the laboratory animal resources facility at the Comparative Medicine and Technology Unit (CoMeT), UPM complying to the standard condition outlined by the UPM ethics committee's guidelines for the care of laboratory animals. All mice were handled in humane and ethical manner and were housed under 12-h dark-light cycle with an ambient temperature regulated at ~28 ± 2 °C. Sterilized or disinfected tap-water and standard pellet diet were provided daily throughout the study period. There was no enrichment provided throughout the study. Changing of bedding and litter tray was carried out twice a week. Overdose of ketamine and xylazine (200 mg/kg and 200 mg/mL per kg body weight, respectively) was used to euthanise the mice to avoid or limit pain/distress in the animal. The mice were also euthanised if,

(1) severe body weight loss up to 10% in 1 week; (2) animal showing no inclination to feed or drink; (3) tumor in mice appear to have a rupture; and (4) tumor size exceeds the allowed range (>2 cm). However, no mice were euthanised prior to the planned end of the experiment. Any surviving mice at the conclusion of the experiment were euthanized. This study was approved by the International Animal Care and Use Committee, UPM (Reference Number: UPM/IACUC/AUP/RO63/2017) and the experiments were conducted based on the approved guidelines.

## Cancer cell preparation and injection into mice

The HT29 human colon carcinoma cell line was used in this study and was harvested from 70% confluent cell cultures. Cell suspensions were prepared in PBS to give approximately $1 \times 10^7$ cells/mL for injection. To generate xenograft, 100 µL of the HT29 cell suspension was injected at the subcutaneous site of the left hind leg of the NCr-Foxn1nu nude mice using a 27-gauge needle (Teruma, Somerset, NJ, USA). The mice were observed daily until tumor masses develop ($\pm$50 mm$^3$).

## Treatment of the tumor-challenged nude mice

The nude mice were grouped into four study groups, comprising one group of non-tumor bearing mice (Normal) and three groups of tumor-bearing mice. The three tumor-bearing mice groups were treated with 0.1 mL of PBS (Negative control), AF2240-i (1280 HA/mL), and rAF-IL12 (1280 HA/mL) respectively via intra-tumoral injections, twice a week for four cycles (i.e., eight times in a 28-day treatment).

## Measurement of tumor growth

Tumor size was estimated by measuring the greatest longitudinal diameter (length) and the greatest transverse diameter (width) using a Vernier calliper. Tumor volume was estimated using the standard formula by *Kersemans et al. (2013)*:

Tumor volume (mm$^3$) = $0.5 \times$ length $\times$ (width)$^2$

## Tissue collection

Whole blood and serum samples were collected for further analysis, while tissue samples such as tumors and vital organs (lung, spleen, liver, and kidney) were also harvested. Each tissue sample was cut into two halves, with one half placed in tube containing 10% buffered formalin for fixation and histopathology analysis and the other half placed in RNAlater solution (ThermoFisher Scientific, USA) and stored in −80 °C freezer for subsequent molecular analysis.

### Hematoxylin and eosin histopathology staining

The organs for histopathological examination were fixed in 10% buffered formalin and embedded in paraffin. For Hematoxylin and eosin (H&E) staining, the tissue was subjected to deparaffinization using xylene followed by rehydration using decreasing concentrations of ethanol (i.e., 100%, 90%, 80% and 70%) before rinsing with tap water for 5 mins. The slides were then stained with the Harris Hematoxylin and counterstained with eosin.

Finally, the slides were mounted with mounting media before observation under a bright-field microscope (Nikon, Japan) (*Cardiff, Miller & Munn, 2014*).

### Serum biochemical analysis

To verify the status of liver and kidney function, concentration of enzymes and biomarker activities such as aspartate aminotransferase (AST), alkaline phosphates (ALP), alanine aminotransferase (ALT), and creatinine were measured. Serum from blood was analysed for its level of biochemical properties using the standard assay kits (Roche Diagnostic GmbH, USA), following the manufacturer's protocol (*Abu et al., 2018*).

### Serum detection of IL-2, IL-12 and IFN-γ cytokines

The expression of IL-2, IL-12 and IFN-γ in mouse serum was measured using the quantitative enzyme-linked immunosorbent assay (ELISA) kits purchased from Biolegend, USA, following the manufacturer's protocol. Colorimetric analysis was conducted by measuring absorbance at 450 nm and 570 nm wavelengths using the μQuant ELISA microplate reader (Bio-Tek Instruments, USA).

### NanoString gene expression analysis

The gene expression analysis of tumor RNA was conducted according to the manufacturer's guide of the nanoString nCounter TagSet Elements™. A volume of 8 μL of master mix (TagSet, hybridization buffer, 0.6 nM probe A, and 3 nM probe B) was transferred into the hybridization/strip tubes and 7 μL of RNA sample was added. Hybridization was conducted by incubating the mixtures at 67 °C for 16 h in a thermal cycler. The samples were then, inserted into the Prep Station machine for purification and immobilization onto the internal surface of a sample cartridge for about 2 h before being transferred to a Digital Analyzer machine for imaging and analysis of gene expression.

### TUNEL assay

The degree of apoptosis induction of treatments was determined using the DeadEnd™ colorimetric TUNEL System (Promega, USA) according to the manufacturer's protocol. The paraffin was removed from the embedded tumor-sectioned slides by immersing them into xylene, followed by rehydration using graded ethanol (100%, 95%, 85%, 70% and 50%), and fixation using 4% paraformaldehyde. Tissue sections were subsequently permeabilized using Proteinase K and equilibrated using equilibration buffer. Fragmented DNA was labelled by incubation with rTdT mixture and 2x SSC termination solvent was used to terminate the reaction. The slides were then, incubated with streptavidin HRP followed by incubation with substrate solution (DAB) for colorimetric detection. The slides were mounted with glycerol and examined under bright-field inverted microscope (Nikon, Japan) (*Ben-Izhak et al., 2007*).

### Statistical analysis

All experiments were evaluated using one-way ANOVA on the GraphPad Prism seven software (GraphPad Software Inc., La Jolla, CA, USA). All in vitro and in vivo tests were

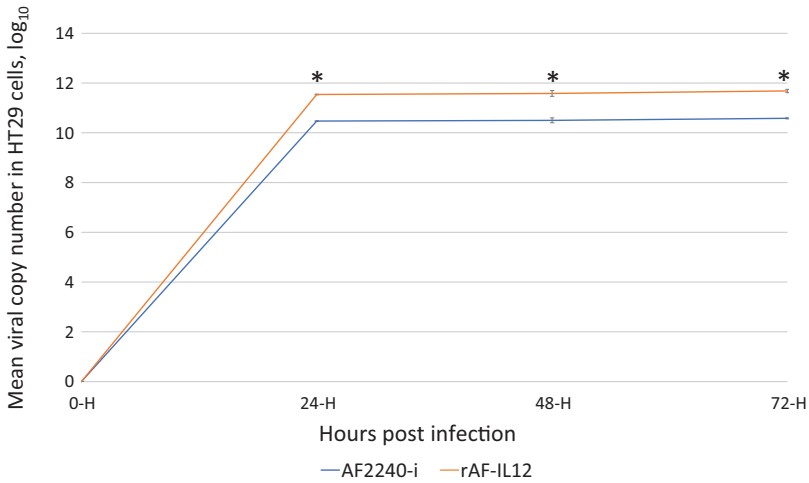

**Figure 1 Growth kinetics curve of AF2240-i and rAF-IL12 based on the viral copy number of these viruses detected in HT29 cells at 24, 48 and 72 h post-infection as measured by RT-qPCR analysis.** The copy number was calculated based on the formula generated from the qPCR standard curve of the NDV: $X = (y - 58.149)/-3.371$; where $X$ is the viral copy number; $y$ is the value mean Cq; 58.149 is the $y$-intercept value; and $-3.371$ is the slope of the standard curve. Data are presented as mean $\pm$S.E.M from triplicate determinations. Statistically significant differences between the means were determined by one-way ANOVA followed by Duncan post hoc test. Differences were considered significant when the *$p \leq 0.05$.

carried out in three replicates and the results are expressed as mean ± standard error of mean (S.E.M.). Significance values of $p < 0.05$ were considered as statistically significant.

## RESULTS

### Viral replication kinetics of rAF-IL12 inside HT29 cancer cells

Viral replication/growth kinetics of rAF-IL12 in HT29 cells was assessed by quantifying viral copy number through TaqMan real-time PCR using total RNA extracted from infected HT29 cells at 24, 48 and 72 hpi. Based on Fig. 1, the rAF-IL12 showed significantly ($p < 0.05$) higher viral copy number in HT29, indicating higher replication/growth kinetics in HT29 cells, compared to AF2240-i. This indicated that the incorporation of IL-12 gene into the AF2240-i anti-genome did not disrupt the resulting ability of the recombinant rAF-IL12 to replicate in neoplastic cells.

### TCID$_{50}$ of viruses in HT29 cells

Table 1 represents data of TCID$_{50}$ of the AF2240-i and rAF-IL12 viruses in HT29 cancer cells. From these data, the infectivity dose/titre of the AF2240-i and rAF-IL12 were $3.16 \times 10^4$ TCID$_{50}$/mL and $4.68 \times 10^4$ TCID$_{50}$/mL, respectively. The results indicated that the rAF-IL12 had higher number of infectious virus particles per volume when compared to the AF2240-i.

### Induction of apoptosis by rAF-IL12 in vitro

MTT assay was conducted to assess the cytotoxic effects of parental AF2240-i and rAF-IL12 on HT29 colon cancer cell line (Fig. 2). Based on Table 2, infection of HT29 with

**Table 1 EC$_{50}$ (half-maximal inhibitory concentration, HA unit) of AF2240-i and rAF-IL12 in HT29 and 3T3 cells 72-h post-infection.**

| Column number in 96-well plate | Log$_{10}$ titre of virus dilution | Observed cytopathic effect (CPE) percentage (%) | |
| --- | --- | --- | --- |
| | | AF2240-i | rAF-IL12 |
| A | 0 | $4/4 \times 100 = 100$ | $4/4 \times 100 = 100$ |
| B | −1 | $4/4 \times 100 = 100$ | $4/4 \times 100 = 100$ |
| C | −2 | $4/4 \times 100 = 100$ | $4/4 \times 100 = 100$ |
| D | −3 | $4/4 \times 100 = 100$ | $4/4 \times 100 = 100$ |
| E | −4 | $0/4 \times 100 = 0$ | $1/4 \times 100 = 25$ |
| F | −5 | $0/4 \times 100 = 0$ | $0/4 \times 100 = 0$ |
| G | −6 | $0/4 \times 100 = 0$ | $0/4 \times 100 = 0$ |
| H | No virus (negative control) | $0/4 \times 100 = 0$ | $0/4 \times 100 = 0$ |

**Note:**

The TCID$_{50}$ was calculated based on the Reed and Muench method. The difference of algorithm of AF2240-i-infected cells = $(100 − 50)/(100 − 0) = 0.50$; log$_{10}$ 50% end-point dilution = $−3 − (0.50) = −3.50$; 50% end-point dilution = $10^{−3.50}$; the titre of the virus: $10^{3.50}$ TCID$_{50}$/0.1 mL = $3.16 \times 10^4$ TCID$_{50}$/mL. The difference of algorithm of rAF-IL12-infected cells = $(100 − 50)/(100 − 25) = 0.67$; log$_{10}$ 50% end-point dilution = $−3 − (0.67) = −3.67$; 50% end-point dilution = $10^{−3.67}$; the titre of the virus: $10^{3.67}$ TCID$_{50}$/0.1 mL = $4.68 \times 10^4$ TCID$_{50}$/mL. The TCID$_{50}$ was presented as mean from four replicate determinations.

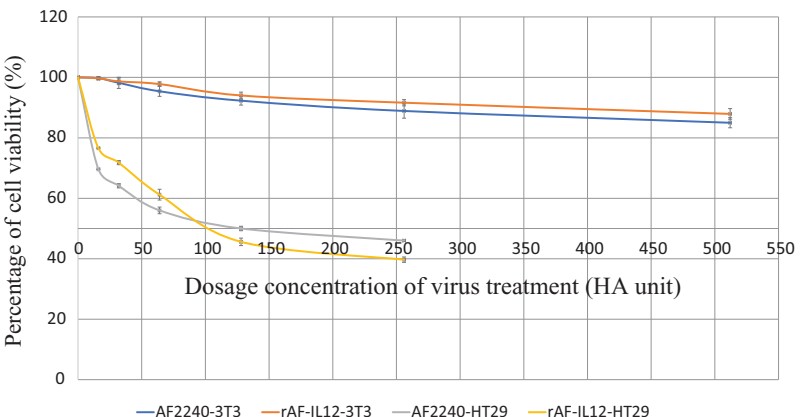

**Figure 2 MTT assay showing the cytotoxicity activity of AF2240-i and rAF-IL12 against HT29 and 3T3 cells 72-h post-infection.** Percentage of viability of two cell lines (HT29 and 3T3) when treated with seven doses of virus treatment (AF2240-i or rAF-IL12) after 72 h. The EC$_{50}$ value (half-maximal inhibitory concentration, HA unit) for AF2240-i-HT29 = 128 ± 1.16; rAF-IL12-HT29 = 110 ± 0.58; AF2240-i-3T3 = n.d.; rAF-IL12-3T3 = n.d. Data are presented as mean ±S.E.M from triplicate determinations.

rAF-IL12 resulted in lower EC$_{50}$ compared to infection with the parental AF2240-i (EC$_{50}$ = 110 ± 0.58 HA unit vs. EC$_{50}$ = 128 ± 1.16 HA unit), indicating that rAF-IL12 showed significantly ($p < 0.05$) better anti-proliferative effect against HT29 cells compared to AF2240-i. Nevertheless, both AF2240-i and rAF-IL12 did not inhibit the growth of normal fibroblast 3T3 cells. The ability of rAF-IL12 to cause cancer cell death or apoptosis was further examined by Annexin V/FITC staining assay and flow cytometry analysis. Figure 3 showed the results of Annexin V/FITC staining of the virus infected HT29 cells. There was a cell population percentage shifting from viable cells to early apoptotic to late apoptotic cells (Figs. 3A–3C). The percentage of early apoptotic cells increased from

**Table 2  ALP, AST, ALT and creatinine serum biochemistry profiles of normal, negative control, AF2240-i-, and rAF-IL12-treated colon cancer-challenged mice after 28-days of treatments.**

| Cell line | Virus treatment | $EC_{50}$ (half-maximal inhibitory concentration, HA unit) |
| --- | --- | --- |
| HT29 | AF2240-i | 128 ± 1.16 |
| | rAF-IL12 | 110 ± 0.58* |
| 3T3 | AF2240-i | N/A |
| | rAF-IL12 | N/A |

**Note:**

* Data are presented as mean ±S.E.M from triplicate determinations. Statistically significant differences between the means at $p < 0.05$.

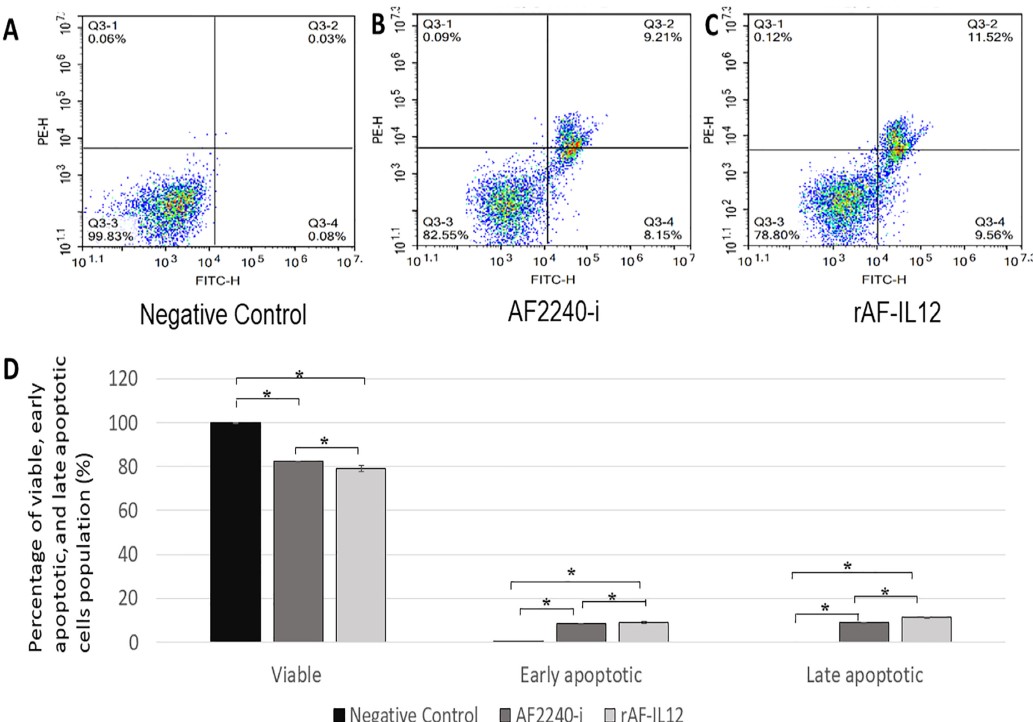

**Figure 3  Annexin V/FITC assay in HT29 cells following AF2240-i (128 HA titre) and rAF-IL12 (110 HA titre) 72 h post-infection.** (A–C) Typical quadrant analysis of Annexin V/FITC flow cytometry of HT29 cells apoptosis. The lower left quadrant of each group indicated the viable cells population; the lower right quadrant indicated the early apoptotic cells population; the upper right quadrant indicated the late apoptotic cells population; and the upper left quadrant indicated the necrotic cells population. Two fluorescent dyes were used in this assay which are FITC (x-axis) and PE (y-axis). (D) Percentage of viable, early apoptotic, and late apoptotic cells population analysed by quantitative analysis. Data are presented as mean ±S.E.M from triplicate determinations. Mean values with statistical difference at $p < 0.05$ between control, AF2240-i and rAF-IL12 are indicated with *.

0.06 ± 0.01% in the negative control group to 8.34 ± 0.14% in AF2240-i-treated cells and 9.03 ± 0.69% in rAF-IL12-treated cells. The percentage of late apoptotic cells was even higher in both AF2240-i- and rAF-IL12-treated cells (9.08 ± 0.09% and 11.20 ± 0.40%, respectively) whereas no cell population entered the late apoptotic phase among the

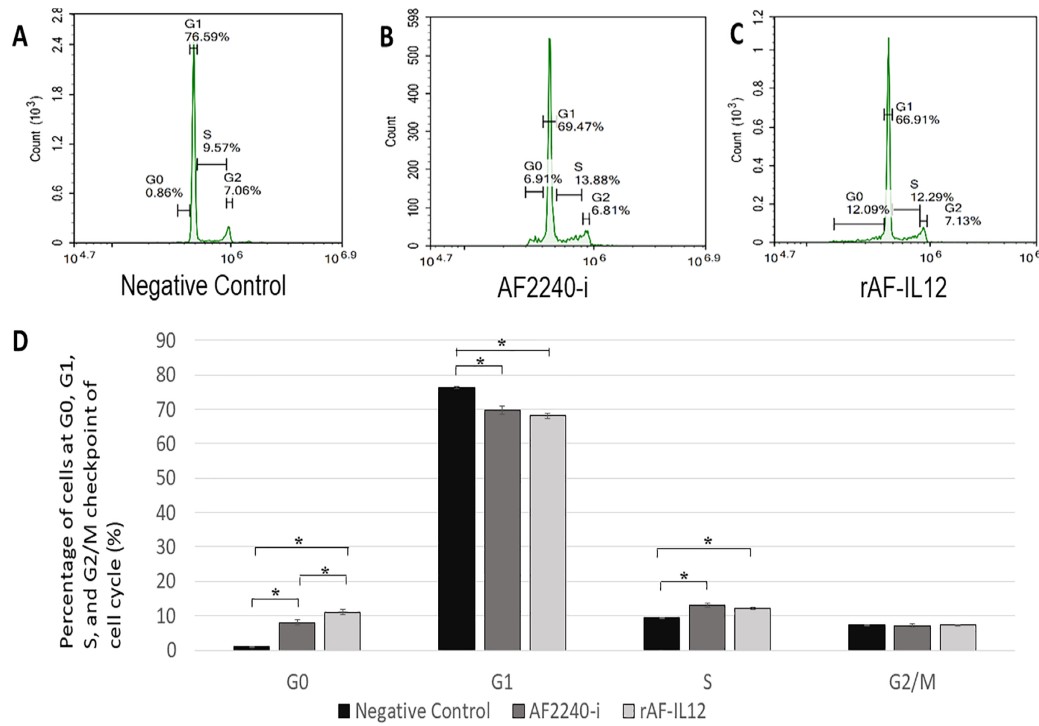

**Figure 4 Cell cycle analysis of HT29 cells at different cell cycle phase.** (A–C) Histogram of cell cycle analysis showing distribution of HT29 cells at different cell cycle phase ($G_0$, $G_1$, S and $G_2$) after 72 h period of treatment with AF2240-i and rAF-IL12. The $G_0$ peak appeared first in the histogram followed by $G_1$, S and $G_2$ peaks indicating the percentage of cells population in those aforementioned cell cycle phases. Percentage of cells population in each peak was calculated from a total number of 10,000 cells in each flow cytometry run. (D) Percentage of cells population at different cell cycle phase analysed by flow cytometer following treatment with AF2240-i (128 HA titre) and rAF-IL12 (110 HA titre) in HT29 cells. Data are presented as mean ±S.E.M from triplicate determinations. Mean values with statistical difference at $p < 0.05$ between control, AF2240-i and rAF-IL12 are indicated with an asterisk (*).

negative control group. To investigate the effects of AF2240-i and rAF-IL12 on the cell cycle progression in HT29 cells, the FACS cell cycle analysis was performed. As shown in Fig. 4, the percentage of HT29 cells at the $G_0$ phase increased significantly from $0.97 \pm 0.11\%$ in the negative control group to $8.08 \pm 0.85\%$ and $11.07 \pm 0.73\%$ in AF2240-i- and rAF-IL12-treated groups, respectively whereas a substantial decrease of the percentage of cells from $76.26 \pm 0.26\%$ in the negative control group to $69.71 \pm 1.26\%$ (AF2240-i) and $68.06 \pm 0.82$ (rAF-IL12) was observed at the $G_1$ phase.

## rAF-IL12 inhibited the growth of HT29 tumor

The therapeutic effects of the rAF-IL12 treatment in nude mice bearing the HT29-induced tumors were assessed after 28 days of treatment. Based on Fig. 5A, the size of tumor in rAF-IL12-treated group was observed to be the smallest compared to negative control and AF2240-i treated group. As shown in Fig. 5B, the line graph plot of tumor volume growth throughout the 28 days post treatment, revealed that the growth of tumor treated with

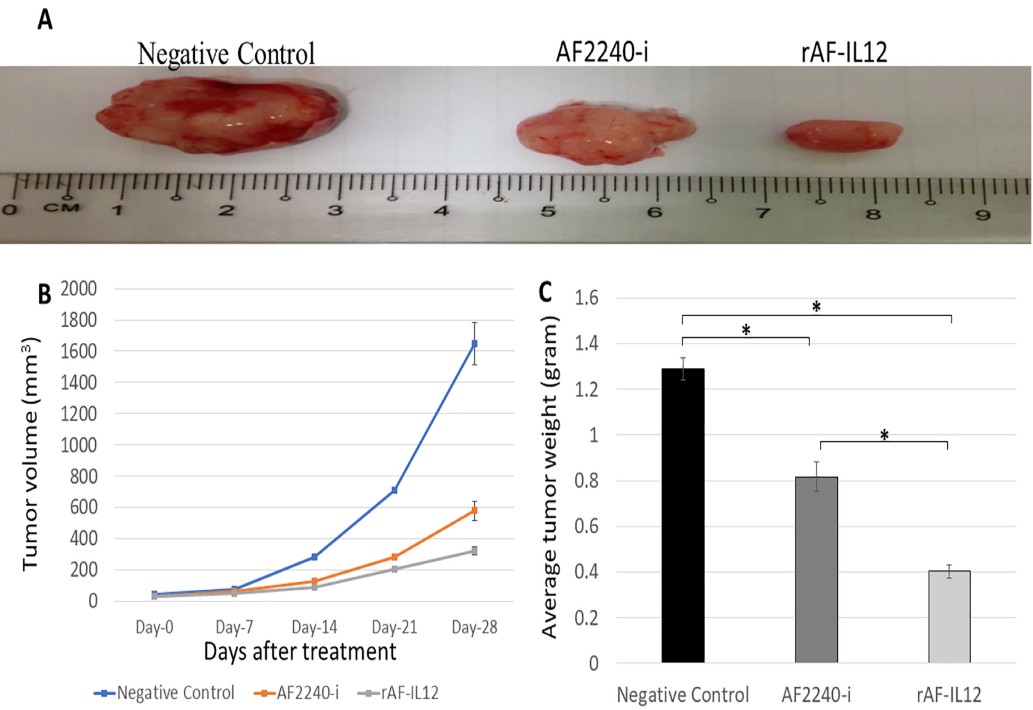

**Figure 5 Effects on the HT29 tumours for the negative control, AF2240-i, and rAF-IL12 groups.**
(A) Images of HT29 tumour harvested from negative control, AF2240-i, and rAF-IL12 groups following the 28-days of treatment. (B) The growth rate of the HT29 tumours from day-0 until day-28 of treatments. (C) Average weight of HT29 tumours harvested from mice after 28-days of treatment. Data are represented as mean ± S.E.M. Mean values with statistical difference at $p < 0.05$ between control, AF2240-i and rAF-IL12 are indicated with an asterisk (*). 

rAF-IL12 was the most restricted compared to the AF2240-i-treated and untreated groups. The final mean tumor volume of mice treated with rAF-IL12 was the smallest ($321.54 \pm 22.90$ mm$^3$) when compared to the negative control ($1649.38 \pm 138.10$ mm$^3$) and the AF2240-i-treated ($577.87 \pm 60.96$ mm$^3$) groups. Based on Fig. 5C, the rAF-IL12-treated group showed the least tumor weight ($0.40 \pm 0.03$ g) compared to the negative control ($1.29 \pm 0.05$ g) and the AF2240-i-treated ($0.82 \pm 0.06$ g) groups. These results indicated that rAF-IL12 treatment of HT29 tumor mass resulted in the slowest growth rate and the lightest tumor weight compared to AF2240-i treatment and the negative control.

## rAF-IL12 replicated inside tumor, lung and spleen

The capability of rAF-IL12 in replicating inside tumor and organs tissues was evaluated using TaqMan real-time PCR through amplification of the NDV F gene. As shown in Fig. 6, the rAF-IL12-treated group showed higher viral copy number in tumor and lung ($9.36$ (log$_{10}$) and $6.42$ (log$_{10}$), respectively) compared to the AF2240-i-treated group ($8.97$ (log$_{10}$) and $6.16$ (log$_{10}$), respectively). On the other hand, the AF2240-i-treated group showed higher viral copy number in spleen ($6.74$ (log$_{10}$)) compared to the rAF-IL12-treated group ($5.47$ (log$_{10}$)).

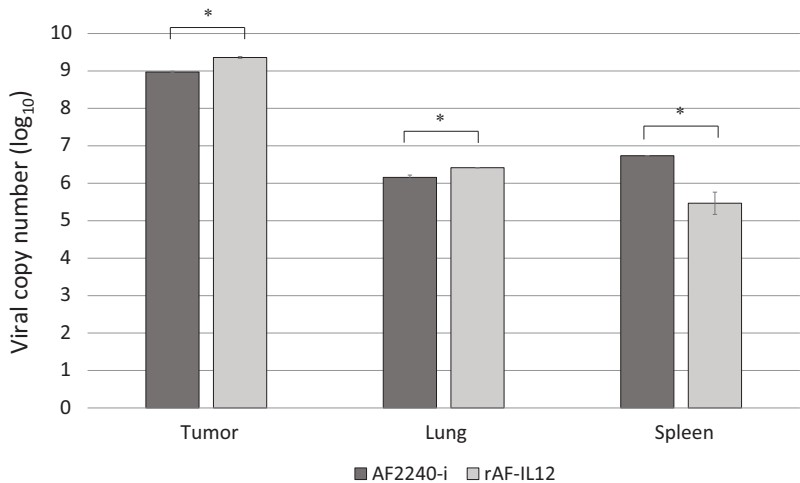

**Figure 6 Viral copy number inside the tumour, lung and spleen (at day-28) of the AF2240-i-treated and rAF-IL12-treated groups of the HT29 colon cancer-challenged mice study as determined by real-time PCR analysis.** Data are presented as mean ±S.E.M from triplicate determinations. Statistically significant differences between the means were determined by one-way ANOVA followed by Duncan post hoc test. Differences were considered significant when the $^*p \leq 0.05$.

## Histopathological safety assessment of organs (lung, spleen, kidney and liver)

Figure 7 shows a photomicrograph of the lungs stained with H&E from the normal, untreated, and treatment groups (AF2240-i and rAF-IL12). Treatment groups and the untreated group had normal alveoli structure comparable to the lungs in the normal group. However, lungs in the AF2240-i-treated group had mild thickening of alveolar interstitial wall due to leucocytic infiltration. This indicated that treatment with rAF-IL12 did not affect the lung structure and function. Figure 8 shows a photomicrograph of histopathological assessment of the spleen from the normal, untreated, and treatment groups (AF2240-i and rAF-IL12). AF2240-i- and rAF-IL12-treated samples showed a normal architecture of spleen with distinct separation of white pulp and red pulp structure similar to spleen in the normal nude mice group. However, depletion of lymphocyte due to degeneration of white pulp area as well as poor distinction between red pulp and white pulp were observed in the spleen of the untreated group. These observations indicated that the rAF-IL12 and AF2240-i treatment did not cause any abnormalities towards spleen as opposed to the spleen in the untreated group. Photomicrographs of histopathological assessment of the kidney from normal, untreated, and treatment groups (AF2240-i and rAF-IL12) are shown in Fig. 9. HT29 tumor-burdened nude mice treated with AF2240-i and rAF-IL12 showed normal kidney architecture (i.e., comparable to the normal group), without obvious pathological lesions whereas the negative control/untreated group had an abnormal form of renal corpuscle with the size of the Bowman's space appearing to be smaller than the kidney from the other groups. Leucocytic infiltrations were also observed in the kidney interstitial space of the untreated and AF2240-i-treated groups. This indicated that treatment with rAF-IL12 did not affect kidney structure and function when

**LUNG**

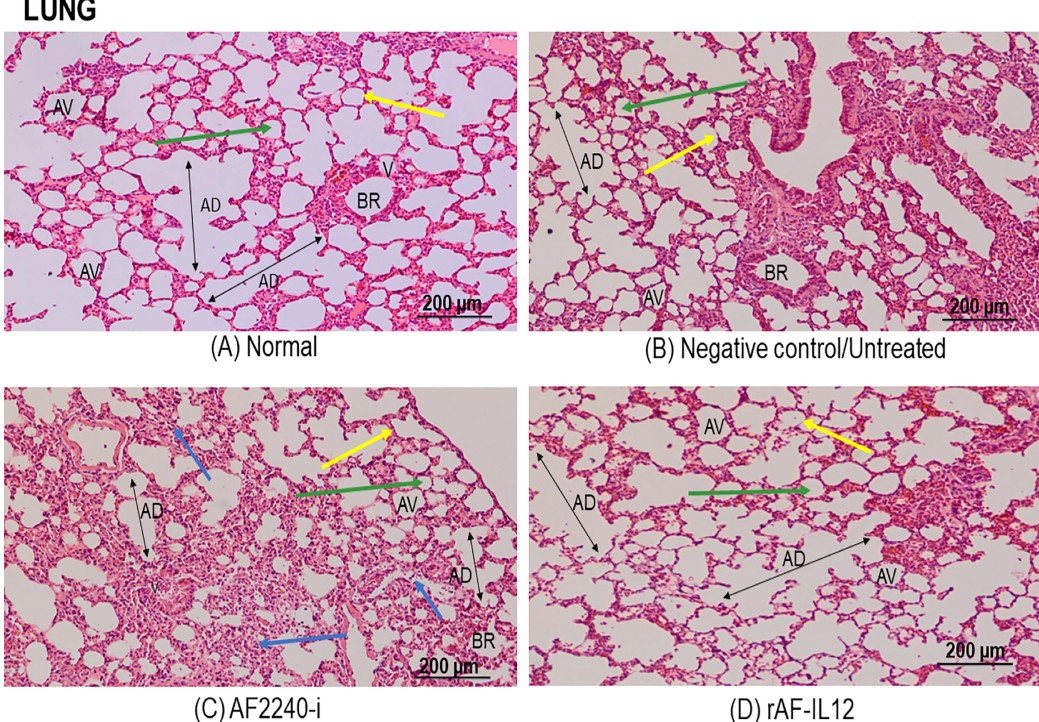

**Figure 7 Photomicrograph section of nude mouse lung stained in H&E.** (A) Normal, (B) Untreated, (C) AF2240-i-treated and (D) rAF-IL12-treated. Normal, untreated and rAF-IL12-treated groups showed normal alveolar morphology; alveolar air space (green arrow) and alveolar capillary (yellow arrow). AF2240-i treated group showed normal alveolar morphology; alveolar air space (green arrow) and alveolar capillary (yellow arrow) but with mild thickening of the alveolar interstitial wall due to leucocytic infiltration (blue arrow). Magnification: 100X; H&E scale bar = 200 μm.

compared to the ones in the untreated group. Histopathological assessment of liver of the HT29 tumor-burden mice treated with AF2240-i and rAF-IL12 as well as the untreated group did not show any obvious degeneration of hepatocytes or liver damage (Fig. 10). However, liver metastasis and leucocytic infiltrations were observed in the untreated and AF2240-i-treated groups. These observations indicated that rAF-IL12 treatment could prevent liver metastasis from occurring in comparison to the AF2240-i-treated and untreated groups.

## rAF-IL12 did not interfere with kidney and liver enzymes function

In this study, ALP, AST, ALT, and creatinine serum levels were evaluated to determine the liver and kidney function following rAF-IL12 treatment on colon cancer-challenged nude mice. Our experiments revealed significant ($p < 0.05$) elevations of ALP, AST, ALT, and creatinine levels in the negative control/untreated tumor-bearing mice group in comparison to the normal group (Table 3). However, there were significant decrease ($p < 0.05$) in ALP, AST, ALT, and creatinine levels in the tumor-bearing mice group receiving the rAF-IL12 treatment compared to the negative control/untreated mice.

**SPLEEN**

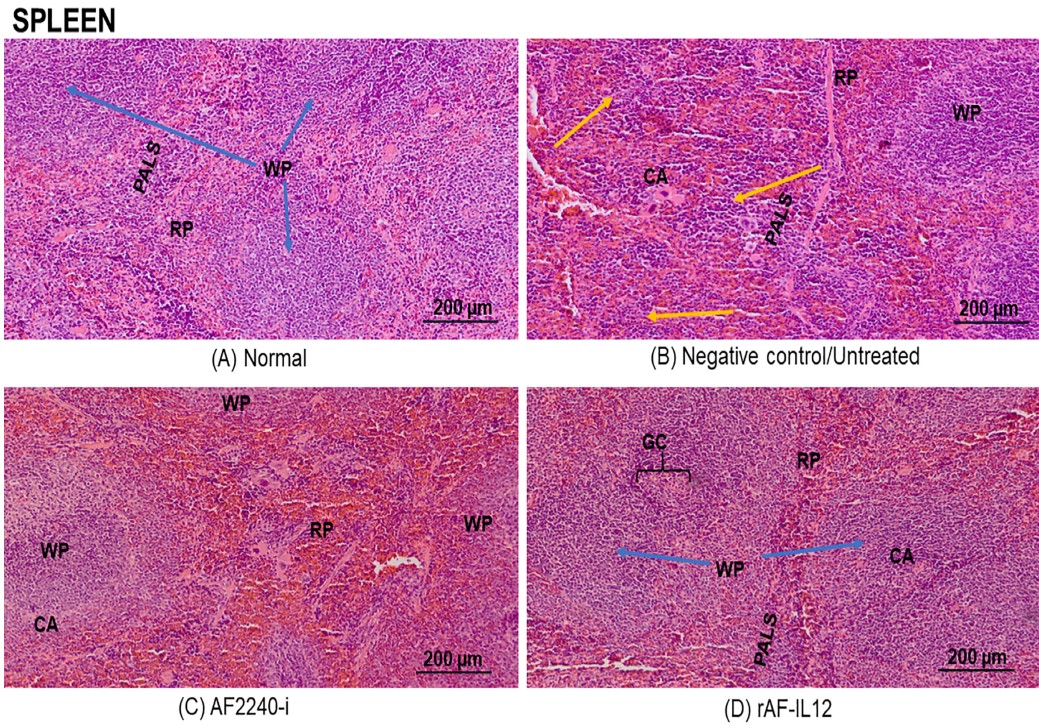

(A) Normal

(B) Negative control/Untreated

(C) AF2240-i

(D) rAF-IL12

**Figure 8 Photomicrograph of the spleen of nude mice stained in H&E.** (A) Normal, (B) Untreated, (C) AF2240-i-treated, and (D) rAF-IL12-treated. Spleen from normal, AF2240-i and rAF-IL12 groups showed no pathological changes with distinct white pulp and red pulp structure. Note lymphocyte depletion (yellow arrow) in the white pulp and poor distinction of the white pulp from the red pulp in the untreated group. WP, white pulp; RP, red pulp; CA, central artery; GC, germinal centre; PALS, peri-arteriolar lymphoid sheaths. Magnification: 100×; H&E scale bar = 200 μm.

Thus, it can be suggested that rAF-IL12 did not cause any adverse effects towards the liver and kidney function and might as well protect those organs from abnormal expression level of serum caused by cancer cells.

## rAF-IL12 modulated the immune system through IL-2, IL-12 and IFN-γ cytokines

Immunomodulation potential of rAF-IL12 in the HT29 tumor-challenged mice was carried out by measuring the expression levels of IL-12, IL-2 and IFN-γ cytokines. Based on Fig. 11A, IL-12 serum cytokine was not detected in all the mice groups except in the tumor-bearing mice receiving rAF-IL12 treatment. A similar pattern was observed in the IFN-γ expression (Fig. 11C). As shown in Fig. 11B, there was a significant increase ($p < 0.05$) of IL-2 level in rAF-IL12-treated mice (21.03 pg/mL) when compared to the negative control/untreated (3.45 pg/mL) and AF2240-i treated mice (15.17 pg/mL), whereas none was detected in the normal mice. Therefore, our results show that rAF-IL12 could regulate all the aforementioned cytokines by increasing their expression levels, which are important in combatting cancer.

**KIDNEY**

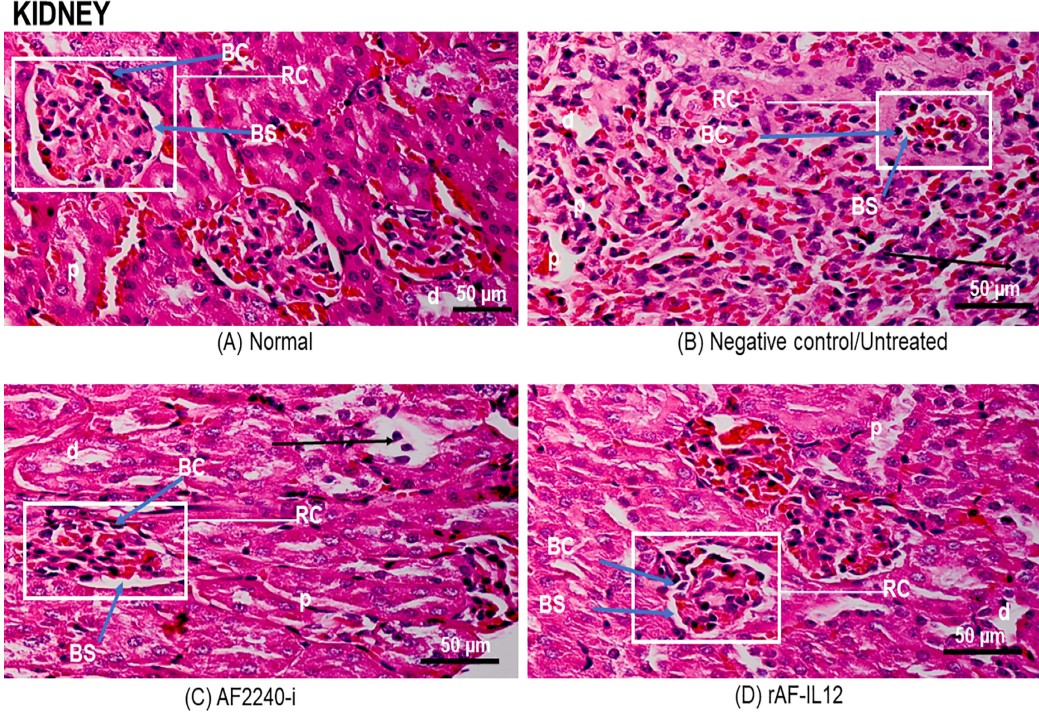

**Figure 9 Photomicrograph section of nude mouse kidney stained with H&E.** (A) Normal, (B) Untreated, (C) AF2240-i-treated, and (D) rAF-IL12-treated. Note leucocytic infiltration in the interstitial space (black arrow) in untreated and AF2240-i and the size of Bowman's space became smaller in the untreated group. RC, renal corpuscle with glomeruli; BS, Bowman's space; BC, Bowman's capsule; p, proximal tubule; d, distal tubule. Magnification: 400X; H&E scale bar = 50 μm.

## rAF-IL12 treatment increased the expression level of apoptosis-related genes

NanoString nCounter TagSet Elements™ gene expression analysis was carried out to measure the expression level of pro-apoptotic genes (Fas, caspase-8, BID, BAX, SMAD3 and granzyme B). The expression levels of the genes are shown in Fig. 12, where rAF-IL12 treatment significantly ($p < 0.05$) increased the expression levels of Fas, caspase-8, BID, BAX and SMAD3 in vitro and in vivo when compared with the negative control group. Granzyme B expression level was also significantly ($p < 0.05$) increased in the tumor-burdened nude mice treated with rAF-IL12.

## rAF-IL12 treatment increased the number of apoptotic cells in TUNEL assay and decreased the number of mitotic cells of the H&E stained tumor

In situ chromatin fragmentation/apoptosis event was confirmed by TUNEL staining of the sectioned tumors of the negative control, positive control, AF2240-i-treated, and rAF-IL12-treated groups obtained from the HT29 tumor-bearing nude mice. TUNEL-stained sections of the rAF-IL12-treated group (60.33) showed clusters of numerous dark brown apoptotic nuclei in the tumor comparable to those in the positive

LIVER

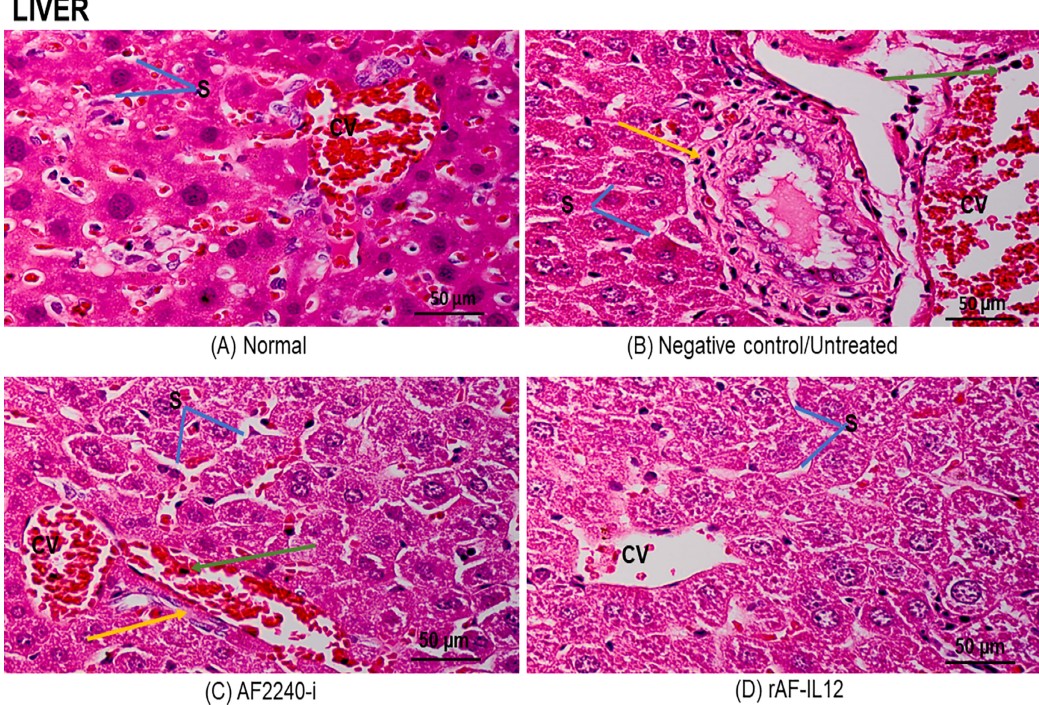

**Figure 10 Photomicrograph of nude mouse liver stained in H&E.** (A) Normal, (B) Untreated, (C) AF2240-i-treated and (D) rAF-IL12-treated. Normal hepatocytes with obvious central vein shown in the normal group and rAF-IL12-treated group. Note the liver metastasis (yellow arrow) and inflammatory infiltrates (green arrow) in the untreated and AF2240-i groups. S, blood sinusoids; CV, central vein. Magnification: 400X; H&E scale bar = 50 μm.

**Table 3 ALP, AST, ALT and creatinine serum biochemistry profiles of normal, negative control, AF2240-i-, and rAF-IL12-treated colon cancer-challenged mice after 28-days of treatments.**

|  | ALP (U/L) | AST (U/L) | ALT (U/L) | Creatinine (Umol/L) |
|---|---|---|---|---|
| Normal | 146.0 ± 1.2[a] | 124.0 ± 0.6[a] | 116.5 ± 0.4[a] | 37.5 ± 0.9[a] |
| Negative control | 187.7 ± 2.2[b] | 190.7 ± 1.2[b] | 148.0 ± 2.3[b] | 48.3 ± 0.3[b] |
| AF2240-i | 166.3 ± 1.8[c] | 146.3 ± 0.3[c] | 129.3 ± 1.5[c] | 40.0 ± 0.6[a] |
| rAF-IL12 | 151.0 ± 0.6[a] | 144.0 ± 0.6[c] | 121.0 ± 0.7[a] | 39.3 ± 1.5[a] |

Note:
Data are presented as mean ±S.E.M from six mice per group. Mean values with different superscript in each column indicate statistical difference ($p \leq 0.05$) between the groups.

control group (i.e., sectioned tumor treated with DNase I) (64.67). The TUNEL-positive apoptotic nuclei were lesser in the AF2240-i-treated group (30.33) while the negative control had the least number of TUNEL-positive apoptotic nuclei appeared in the tumor tissue (7.67) as shown in Fig. 13. Based on Fig. 14, the rAF-IL12 treatment group had the least number of actively dividing cells (i.e., undergoing mitosis) (18.67) compared to the AF2240-i-treated (25.00) and the untreated groups (37.67).

# DISCUSSION

Colorectal cancer (CRC) remains a worldwide burden and is expected to continue to rise to more than 2.2 million new cases and 1.1 million deaths by 2030 (*Arnold et al., 2017*).

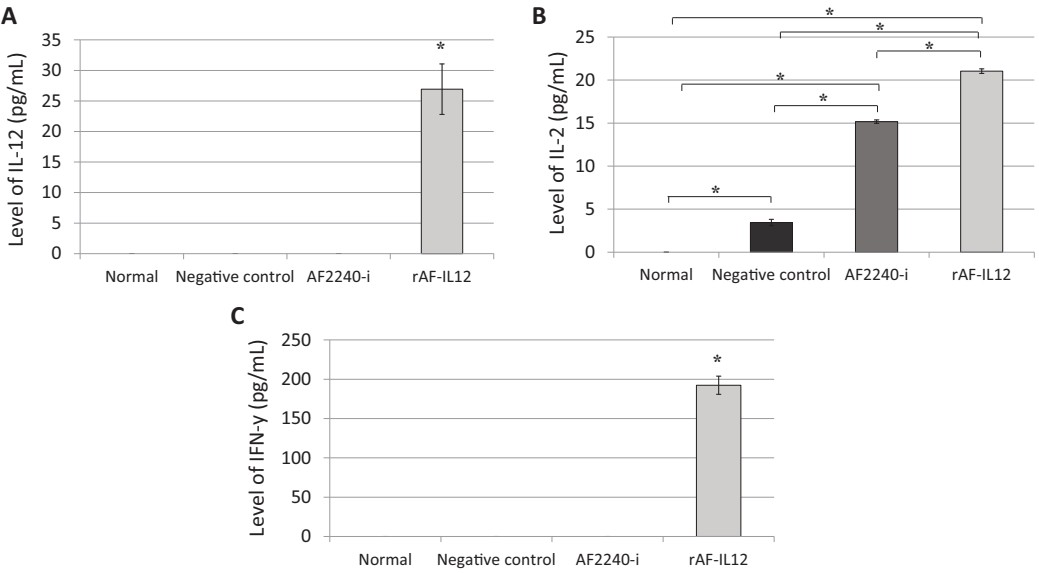

**Figure 11 Serum level of cytokine.** (A) Interleukin-12, (B) interleukin-2 and (C) interferon-γ from normal and colon cancer-challenged mice (negative control, AF2240-i, and rAF-IL12) after 28-days of treatment. Data are presented as mean ±S.E.M. of triplicate determinations. Mean values with statistical difference at $p \leq 0.05$ between groups are indicated with an asterisk (*).

Current chemotherapy and radiotherapy available for colorectal cancer have so far shown limited efficiency due to adverse side effects and emergence of drug resistance among treated patients (*Sveen et al., 2018*). These issues are particularly due to the heterogeneity of colorectal cancer that involves various genetic and epigenetic alterations, resulting in distinct morphological and phenotypic differences among cases that respond differently to therapy (*Gang et al., 2018*; *Merlano et al., 2017*). The HT29 colon cancer cell line has generally been used for the analysis of oncolytic viruses (*Boisgerault et al., 2013*; *Kojima et al., 2010*). It possesses the wild-type KRAS gene and mutated BRAF gene at V600E site, a condition that contributes to resistance towards anti-EGFR therapy such as cetuximab and panitumumab (*Ahmed et al., 2013*; *Linnekamp et al., 2015*; *Zhao et al., 2017*). Other than that, HT29 is classified as a CMS3 subtype CRCs, which is devoid of immune cell infiltration and is also microsatellite stable (MSS), reflecting its low response to immunotherapy (*Guinney et al., 2016*; *Kang et al., 2018*; *Kocarnik, Shiovitz & Phipps, 2015*; *Rozek et al., 2016*).

In this study, the efficacy of the recombinant NDV, rAF-IL12 in targeting the HT29 colon cancer cell was evaluated. The rAF-IL12 was developed by incorporating the IL-12 in the backbone of the wild-type AF2240-i genome to elicit better cancer therapeutic effects. Of note, the infectivity of virus plays a significant role in targeting cancer cells. The $TCID_{50}$ titration method revealed that the rAF-IL12 had higher number of infectious virus particles per volume when compared to the AF2240-i, hence, indicating the higher infectivity possessed by the rAF-IL12 towards HT29 cells. This was in line with the results shown by the TaqMan real-time PCR as the rAF-IL12 showed higher growth/replication kinetics in HT29 cells compared to AF2240-i. Furthermore, the rAF-IL12

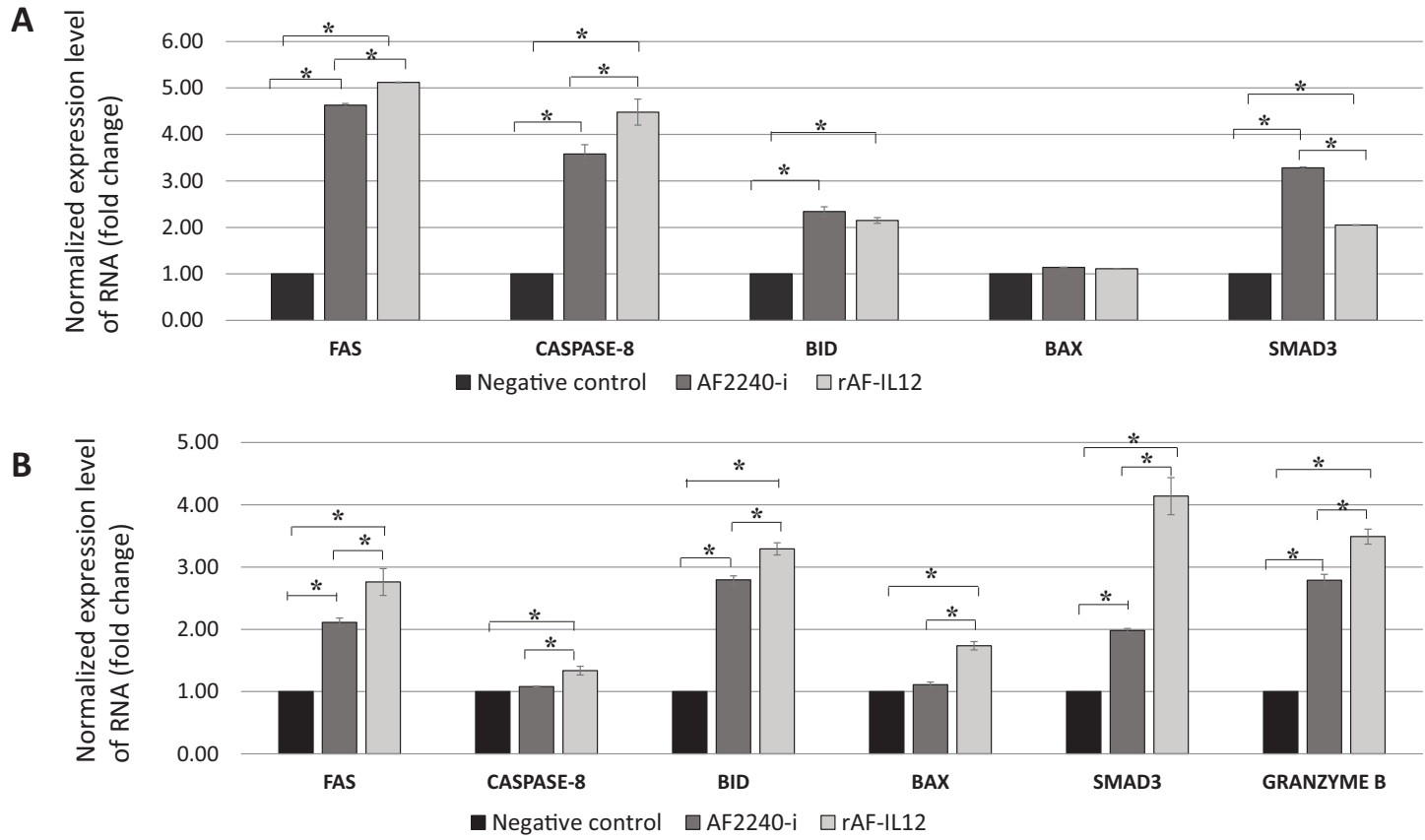

**Figure 12 Normalized gene expression level of Fas, caspase-8, BID, BAX and SMAD3.** (A) HT29 cancer cells in vitro and (B) tumour excised from HT29 tumour-burden mice (note the addition of granzyme B expression). Data are presented as mean ±S.E.M from three independent experiments. Statistically significant differences between the means were determined by One-Way ANOVA followed by Duncan post hoc test. Differences were considered significant when the $p \leq 0.05$ as indicated by an asterisk (*).

showed slightly greater cytotoxicity effects towards HT29 cells when compared to the AF2240-i as shown in the MTT assay and was further confirmed through Annexin V FITC and cell cycle analysis. Interestingly, it is of importance to note that the ability of NDV to selectively replicate and cause death in cancer cells is due to defective anti-viral signaling pathways among tumor cells which makes them highly susceptible to NDV (*Goldufsky et al., 2013*). In the previous study, the rAF-IL12 has been shown to significantly inhibit the proliferation and induce cytotoxicity against breast cancer cell lines; MDA-MB-231 and MCF-7 in vitro when compared to its parental wild-type AF2240-i (*Amin et al., 2019*). Cytotoxicity against colon and breast cancer indicates the capability and potential of rAF-IL12 in targeting other types of cancer as well.

Following the promising results in vitro, further evaluation of rAF-IL12 was carried out in vivo, using the NCr-Foxn1nu nude mice xenografted with HT29 cell line. This study showed that rAF-IL12 significantly ($p < 0.05$) decreased the volume and weight of HT29 tumors when compared with the parental NDV, AF2240-i and the untreated group. This current finding was in line with *Amin et al. (2019)* where the rAF-IL12 treatment resulted in 52% growth inhibition of 4T1 breast tumor while AF2240-i only caused 34.5%

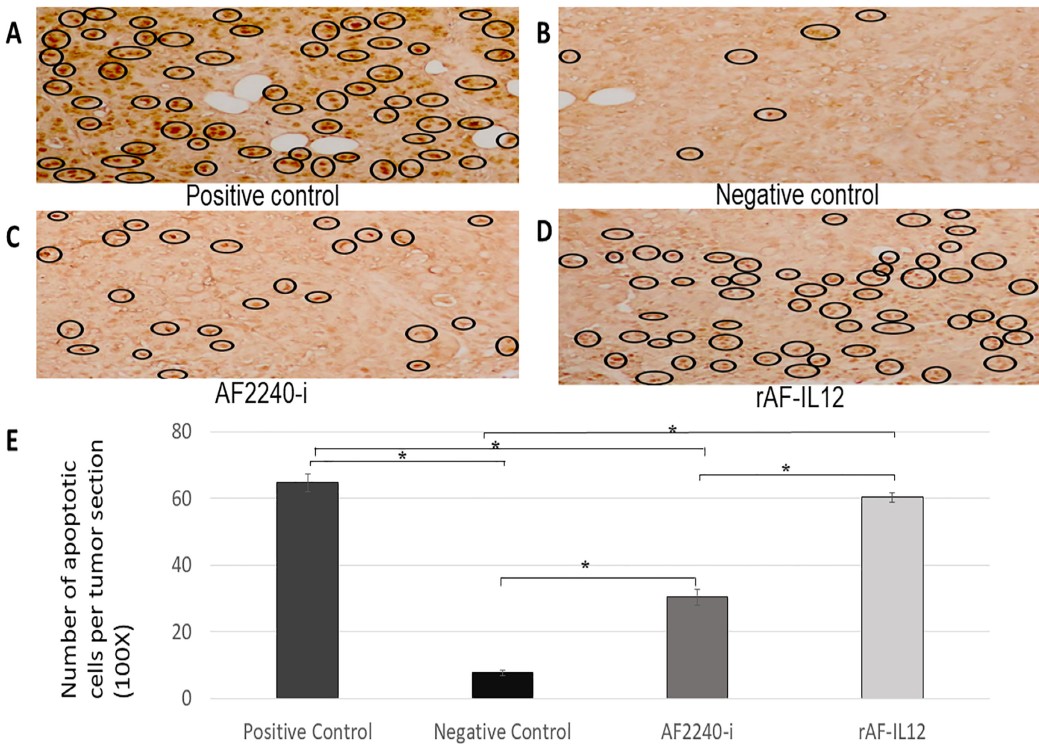

**Figure 13** (A–D) Tumour sections assayed by DeadEnd colorimetric TUNEL system to indicate cell apoptosis in four different groups of HT29 tumour-bearing Balb/c mice; positive control (sample treated with DNAse I), negative control, tumour treated with AF2240-i, and tumour treated with rAF-IL12. Brown stained nuclei in the black circle indicate DNA fragmentation and nuclear condensation. Magnification: 100x; (E) the number of apoptotic cells per tumour section from the four aforementioned groups after 28 days of treatment. Data are presented as mean ±S.E.M. of triplicate determinations. Mean values with statistical difference at $p \leq 0.05$ between groups are indicated with *.

inhibition as compared to the negative control group. This indicates that the rAF-IL12 was far more potent in inhibiting the growth and progression of cancer than its wild-type AF2240-i. However, it is of importance to note that the recombinant NDV, rAF-IL12 was able to inhibit the early stage of tumor growth in mice but the effect on the late stage is not covered in this study and definitely is important to be determined in the future. Other than that, it is also important to determine the possibility of re-growth of the tumor after 28 days when the virus is no longer injected. This study would exhibit the potential ability possessed by the recombinant NDV, rAF-IL12 in preventing the recurrence of colon cancer in the future. In addition to its tumor-suppressing effect, the safety aspects of the usage of rAF-IL12 as a tumor vaccine treatment in colon cancer-challenged mice were also evaluated by quantifying the viral copy number in tumor and vital organs such as lung, spleen, liver, and kidney. Higher viral copy number of rAF-IL12 in solid tumor indicates that rAF-IL12 exhibits higher tumor specific replication compared to its parental strain. Moreover, this result further proves that rAF-IL12 was capable to replicate not just in the cancer cell line but also in the tumor tissue. The intra-tumoral-injected rAF-IL12 and AF2240-i was found to disseminate to lung and spleen with low viral copy

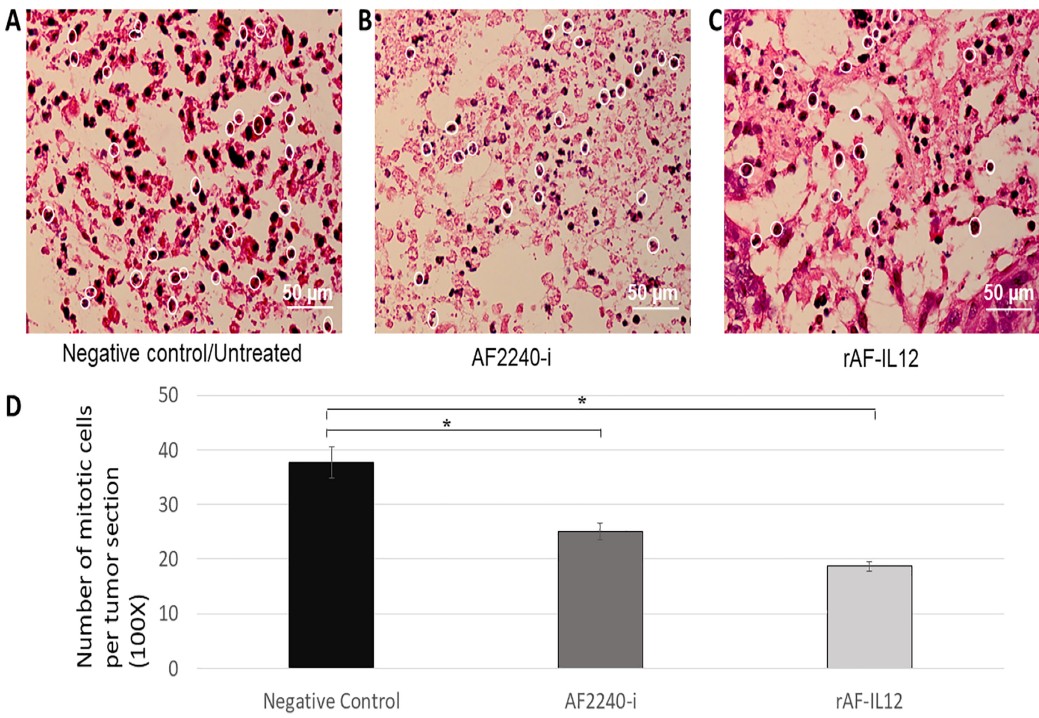

**Figure 14 (A–C) Photomicrograph section of the tumour mass of mice stained with H&E from three different groups of mice, negative control/untreated, AF2240-i-treated, and rAF-IL12-treated. The white circles indicate the actively dividing/mitotic tumour cells in the aforementioned groups. Magnification: 400X; H&E scale bar = 50 µm; (D) The number of mitotic cells per tumour section from the negative control, AF2240-i-treated, and rAF-IL12-treated after 28-days of treatment.** Data are presented as mean ±S.E.M. of triplicate determinations. Mean values with statistical difference at $p \leq 0.05$ between groups are indicated with an asterisk (*).

number detected in both organs, while no viral presence was detected in kidney and liver. This suggests that the use of the viruses had minimal impact towards normal tissues and could be considered safe. Moreover, limited viral dissemination from injected tumor mass into other organs is crucial for effective oncolytic activity, since the disseminated virus would be eliminated by antiviral activities of the interferons, cytokines and infiltrating leukocytes (*Bian et al., 2006*). Furthermore, histopathology analysis of the sectioned lung, spleen, liver, and kidney showed that rAF-IL12 did not cause any obvious pathological lesion. The aforementioned organs are often selected for preliminary toxicity observation in mice (*Jing et al., 2014*; *Zhang et al., 2016*; *Yu, Drisko & Chen, 2013*).

It has been elucidated that malignancies contribute to elevated level of serum biochemistry parameters such as aspartate transaminase (AST), alanine transaminase (ALT), alkaline phosphatase (ALP) and creatinine whereby these enzymes were indicators for liver and kidney injuries (*Chauhan et al., 2016*; *Kim et al., 2008*; *Saif, Alexander & Wicox, 2005*). In this study, all the aforementioned enzymes levels were elevated in the untreated tumor-challenged nude mice in comparison to the level in the normal nude mice (without tumor) group. However, the enzyme levels in the rAF-IL12-treated group were the closest to the normal nude mice group, rendering the intra-tumoral injection of rAF-IL12 to

have very minimal/limited toxicity towards liver and kidney function. Our results are in agreement with a previous study in which the serum biochemistry level of AST and ALT were elevated on day-1 but normalized by day-3 while creatinine levels remained normal at both day-1 and day-3 following the administration of NDV(F3aa)-GFP intra-peritoneally to treat gastric cancer (*Song et al., 2010*).

The rAF-IL12, like any other NDV, could invoke specific host immune response against tumor tissue. In addition to the processed antigen at major histocompatibility complex I (MHC-I), immune cells could also target the viral envelop protein HN on the surface of the rAF-IL12-infected tumor hence, could prevent the cancer from escaping the immune destruction through the downregulation of MHC-I expression as shown in previous studies (*Zakay-Rones, Tayeb & Panet, 2015*; *Schirrmacher, 2017*; *Al-shamery et al., 2011*; *Pan et al., 2013*). Therefore, for a better understanding of a possible immunological effect of rAF-IL12 on HT29 xenografts in nude mice, the level of several important cytokines such as interleukin-12, interleukin-2 and IFN-γ were evaluated. The data revealed that the expression levels of all the tested cytokines were significantly ($p < 0.05$) upregulated in comparison to the untreated group following tumor colonization by the rAF-IL12. Interestingly, despite invoking the same immune response as the parental AF2240-i, the IL-12 expression of rAF-IL12-infected tumor mass could enhance its in vivo oncolytic activity and edging out the AF2240-i as rAF-IL12 treatment was the only group that managed to elicit the expression of IL-12 and IFN-γ cytokines in the nude mice. Additionally, rAF-IL12 increased the expression level of IL-2 by 1.39-fold when compared to the AF2240-i. This could only serve as an evidence for significant difference in suppressive effect of tumor growth between rAF-IL12 and AF2240-i treatment. IL-12 and IFN-γ presence is important for anti-tumoral effect of rAF-IL12 as previous studies demonstrated that IL-12 treatment inhibited lung tumor growth, resulting in the long-term survival of lung cancer-bearing mice while IFN-γ mediated the anti-tumor effects of radiation therapy in murine colon tumor (*Gerber et al., 2013*; *Yue et al., 2016*). IL-12 has also been shown to enhance natural killer (NK)-cell mediated (i.e., innate immunity) cytotoxicity against tumor (*Parihar et al., 2002*), possibly explaining the efficacy of the rAF-IL12 in limiting the progression of HT29 cancer cells in this study (i.e., nude mice lack T-cells). Previous studies have shown that oncolytic adenovirus co-expressing IL-12 and IL-18 led to the infiltration of NK cells into melanoma tumor tissues in murine models, while in another study, the efficacy of Sindbis-virus-based vectors against human ovarian carcinoma xenografts was found to be largely NK-cell-dependent and enhanced by IL-12 arming (*Choi et al., 2011*; *Granot et al., 2011*). Therefore, this suggests that the rAF-IL12 could possibly enhance the infiltration of NK cells and innate immunity as well. Briefly, the fact that recombinant NDV, rAF-IL12 encodes for the IL-12 gene has made it possible for the generation/production of IL-12 cytokine in abundance during the viral replication process hence, further stimulating the immune response machinery towards cancer, something that could not be achieved by the parental AF2240-i. In this regard, the AF2240-i falls behind compared to the rAF-IL12. The increased level of the aforementioned cytokines as a result of rAF-IL12 treatment had successfully reversed the immunosuppressive tumor microenvironment and thereby,

reinstating immunosurveillance, which ultimately resulted in the suppressed tumor growth of HT29 colon cancer-challenged nude mice (*Kalyanasundram et al., 2018*; *Rausalova & Krepela, 2010*).

Severely damaged cells are directed into a programmed cell death through either extrinsic or intrinsic (i.e., mitochondrial) pathway depending on the source of stimuli (*Elmore, 2007*). In this study, the expression level of pro-apoptotic genes such as the Fas receptor, initiator caspase-8, Bid and Bax were elevated suggesting that rAF-IL12 induced apoptosis through extrinsic and intrinsic pathways. In extrinsic pathway, the binding of Fas ligand to Fas cell surface death receptors induces the formation of the death inducing signaling complex (DISC) followed by the activation of caspase-8 that eventually cleaves and activates the effector caspase-3 and -7 to finally result in apoptosis (*Elmore, 2007*). Caspase-8 also provides a molecular link between the extrinsic and intrinsic apoptotic pathways whereby it cleaves the pro-apoptotic Bcl-2 family member, Bid, into a 15 kDa protein, namely, truncated Bid (tBid). In its activated form, tBid heterodimerizes with Bax and subsequently inducing mitochondrial outer membrane permeabilization (MOMP) in cells, which later results in the release of cytochrome c into cytosol. The binding of cytochrome c to cytosolic Apaf-1 results in the formation of apoptosome, which then leads to the activation of caspases and subsequently, cell death (*Kantari & Walczak, 2011*; *Westphal et al., 2011*). Previous studies have shown that the loss of Bax and caspase-8 expression in tumor cells are not only causing evasion of apoptosis but are also correlated with resistance to drug-induced apoptosis treatment as well as giving shorter survival time in cancer patients (*Paul-Samojedny et al., 2005*; *Pryczynicz et al., 2014*; *Sträter et al., 2010*). Additionally, rAF-IL12 was shown to also increase the expression level of Smad3, which is known to be involved in inhibiting survival, proliferation, and tumorigenesis in colon cancer in response to transforming growth factor-β (TGF-β) stimulation. Previous studies have demonstrated that inhibition of Insulin Receptor Substrates-1 (IRS-1) by TGF-β/Smad3 leads to suppression of X-linked inhibitor of apoptosis protein (XIAP) and cyclin D1 that are responsible for inhibiting caspases, leading to apoptosis evasion. Additionally, Smad3 activates the pro-apoptotic protein, Bad, which consequently results in TGF-β-mediated induction of apoptosis (*Bailey et al., 2017*; *Mithani et al., 2004*). Other than that, serine proteinase granzyme B was also significantly up-regulated by rAF-IL12 treatment. Granzyme B together with perforin granules are released by cytotoxic T-lymphocytes and natural killer (NK) cells to eliminate harmful target cells including virally infected, allogenic and transformed cells (*Mhaidat et al., 2014*). In this study, it is plausible to assume that the IL-12 from rAF-IL12 stimulates the production of granzyme B through the NK cells as nude mice lack cytotoxic CD8 T-cells (due to lack of thymus). Perforin disrupts the lipid membrane of the target cells by forming pores, which affects cellular fluidity and content as well as providing the entry for granzyme B (*Li et al., 2014*; *Mhaidat et al., 2014*). Once granzyme B is translocated into the target cell cytoplasm it can initiate cell apoptosis in multiple ways, including direct proteolytic processing and activation of the executioner procaspase-3 and -7, cleaves multiple intracellular housekeeping proteins such as α-tubulin, filamin and β-fodrin, and altering the outer mitochondrial membrane

through conversion and activation of protein Bid to the MOMP-inducing tBid fragment (*Rausalova & Krepela, 2010*). The importance of granzyme B expression is justified by previous study which demonstrated that low expression of granzyme B was associated with early signs of metastasis in CRC while higher tumor infiltration with CTL and granzyme B improved all-cause and cancer specific survival of CRC patients, irrespective of stage (*Prizment et al., 2017*; *Salama et al., 2011*). However, it is also of importance that the expression levels of the aforementioned genes in the normal cell lines or normal mice to be determined as to further prove the recombinant NDV does not cause any apoptosis in non-cancerous cells. The induction of apoptosis by rAF-IL12 could also be observed by the TUNEL assay whereby higher number of DNA fragmentation was observed in the rAF-IL12 treated group as compared to the negative control and AF2240-i treated groups. It is a good indicator as DNA fragmentation is one of the vital features in apoptotic event (*Elmore, 2007*). Further evidence in favor of rAF-IL12 as anti-cancer vaccine was the decreased number of actively dividing cells (mitotic cells) as portrayed by the sectioned tumor tissues stained with hematoxylin and eosin.

## CONCLUSIONS

In conclusion, this study suggested that rAF-IL12 has promising anti-cancer potential especially in treating CMS3 subtype colorectal cancer as it was able to cause cell death in vitro and inhibit the growth of HT29 tumors in vivo while exhibiting a safety profile with no side effect towards normal tissues and organs of HT29 tumor-challenged nude mice. Additionally, rAF-IL12 regulated the immune system of the tumor-burdened nude mice and increased the expression levels of apoptosis-related genes.

## ACKNOWLEDGEMENTS

We thank all the Science Officers and staff at the Laboratory of Vaccine & Immunotherapeutics (LIVES), UPM for their assistance with this study. Our sincere gratitude goes to Dr. Ng Wei Lun for his assistance with proofreading and language editing of this manuscript. We would also like to thank the Attending Veterinarian and staff at the Comparative Medicine and Technology Unit (COMeT), UPM for granting us access to the facility and resources there.

### Funding
This study is supported by MOSTI Flagship Fund, reference number: FP0514B0021-2 (DSTIN). The funders had no role in study design, data collection and analysis, decision to publish, or preparation of the manuscript.

### Grant Disclosures
The following grant information was disclosed by the authors:
MOSTI Flagship Fund: FP0514B0021-2(DSTIN).

## Competing Interests

The authors declare that they have no competing interests.

## Author Contributions

- Syed Umar Faruq Syed Najmuddin performed the experiments, analyzed the data, prepared figures and/or tables, authored or reviewed drafts of the paper, and approved the final draft.
- Zahiah Mohamed Amin performed the experiments, prepared figures and/or tables, and approved the final draft.
- Sheau Wei Tan performed the experiments, analyzed the data, prepared figures and/or tables, and approved the final draft.
- Swee Keong Yeap conceived and designed the experiments, analyzed the data, prepared figures and/or tables, and approved the final draft.
- Jeevanathan Kalyanasundram performed the experiments, authored or reviewed drafts of the paper, and approved the final draft.
- Abhimanyu Veerakumarasivam analyzed the data, authored or reviewed drafts of the paper, and approved the final draft.
- Soon Choy Chan analyzed the data, authored or reviewed drafts of the paper, and approved the final draft.
- Suet Lin Chia analyzed the data, authored or reviewed drafts of the paper, and approved the final draft.
- Khatijah Yusoff conceived and designed the experiments, analyzed the data, authored or reviewed drafts of the paper, and approved the final draft.
- Noorjahan Banu Alitheen analyzed the data, authored or reviewed drafts of the paper, and approved the final draft.

## Animal Ethics

The following information was supplied relating to ethical approvals (i.e., approving body and any reference numbers):

This study was approved by the International Animal Care and Use Committee, UPM (Reference Number: UPM/IACUC/AUP/RO63/2017) and the experimentation was conducted based on the approved guidelines.

## Data Availability

The raw measurements are available as a Supplemental File.

## Supplemental Information

Supplemental information for this article can be found online at http://dx.doi.org/10.7717/peerj.9761#supplemental-information.

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
