# Peer review of "Oncolytic effects of the recombinant Newcastle disease virus, rAF-IL12, against colon cancer cells in vitro and in tumor-challenged NCr-Foxn1nu nude mice"

_PeerJ, doi:10.7717/peerj.9761_

## Round 0.1 · original submission · Major Revisions

Both the reviewers have carefully gone through your manuscript and have asked for several clarifications and suggested additional experiments including controls. If you decide to resubmit, please address each of those comments.

Reviewer 1 ·

Basic reporting

The manuscript entitled “Oncolytic Effects of the Recombinant Newcastle Disease Virus Strain AF2240-i Expressing Interleukin-12, rAF-IL12, against HT29 Colon Cancer Cells In Vitro and in Tumor-challenged NCr-Foxn1nu Nude Mice (#47395)” by Najmuddin et al, describes the effect of recombinant NDV that expresses IL-12 on the colon cancer cell line as well as tumor-challenged nude mice. The recombinant virus is replicating well in the cancerous cells in a stable manner. The recombinant virus induces apoptosis in vitro and also reduces tumor growth in vivo without any deleterious effect on normal cells. The authors also described the safety assessment of the recombinant virus for various organs including lung, liver, spleen and kidney. The effect on the cytokine’s modulation along with the increase in the expression of apoptosis-related genes has been observed. Overall this is an interesting paper. The experiments are carefully executed and well described. However, the manuscript needs some improvements.

1. The title seems too descriptive. It can be improved by removing the names of virus strain, cell line.
2. The abstract needs to be modified for better organization of sentences. For example, in line 41, it is difficult to differentiate for the readers whether it is about previous background study or the authors started describing current one.
3. The discussion section seems a bit unorganized too. At some places it looks like a repetition of the results (for example line 416-456), while at others, a lot of literature has been described without connecting the literature to the current study. For instance, line 477: the paragraph starts with “It has been postulated that there are two major modes of oncolysis taken by NDV; one is by selective infection and killing tumor cells, while the other is by indirectly eliciting/invoking specific host immune responses against the tumor tissue”. However, no further connection of these two mechanisms has been discussed later. It should be discussed which possible mode of oncolysis authors find in the current study.
4. The authors described the literature in introduction in details. However, the recombinant NDV studies are not mentioned that used various cytokines expressing virus.

Experimental design

1. The exact same recombinant virus has already been shown as an inhibitor of breast cancer. This is another study of different type of cancer. Is it possible that this recombinant virus has the capability to reduce a number of different kind of tumors? Also, the research has been cited only for the construction of the recombinant virus. More details should also be included in abstract, introduction and discussion.

2. Several studies are available showing the tumor inhibitory effect of recombinant NDV expressing different cytokines. Was there any specific reason for the use of IL-12 among other cytokines? Is IL-12 better for the tumor inhibition as compared to other cytokines?

3. An important study of the survival rate of cancerous mice in untreated and treated groups is missing here.

4. The study shows the inhibition of tumor growth in mouse when treated with recombinant NDV (Figure 6). Does this or any other study have evidence of the reduction in tumor growth after treating mice when the tumor is fully grown. To make it a strategy for the treatment of cancers, the studies should be there which proves the treatment after tumor development as in some cases it is hard to identify individuals before disease progression. For example, what happens if the mice are treated after 14, 21 or 28 days after tumor cells injection. Will they survive?

5. Figure 1: The authors showed the viral replication using qPCR for the quantitation of viral RNA. The growth kinetics should have been performed for the calculation of infectious virus particles.

Validity of the findings

1. Lines 295 and 301: Figures 3 and 4 are showing different results for the similar effect on apoptotic cell population in control, virus treated and recombinant virus-IL-12 treated cells. These both are exact same experiment; the only difference is the use of two different techniques. A result is confirmed only when same results are obtained by using different methods. I do not see any reason for different results here when only the methodology is different. 20 % cells are viable according to figure 3 while 80% cells are viable with different method used in figure 4. This does not look like because of the variability within the experiment as the error bars are too small. How does authors explain this?

2. Line 283: The authors has described the difference between the EC50 values of two viruses as significant. Though these values look pretty similar and it can be due to the variability between two different experiments. Also, in Fig 2, error bars are missing so it is hard to interpret the variations in the cell viability for three independent experiments.

3. Line 327: The authors mentioned that the virus is disseminated in vital organs. Lung and spleen show a significant amount of viral copy number though. Is it possible for the authors to check the infectious virus particles for these organs as well?

Additional comments

1. Line 372: Both viruses show different effects on the cytokine levels. Is it possible that they follow two different mechanisms/pathways to induce apoptosis?
2. Line 459: “detected in kidney and lung”. Authors did not detect viral RNA in kidney and liver.
3. Figure 3: How many cells were calculated in total?
4. Line 470: “However, the enzyme levels were restored back almost to their normal level in the rAF-IL12-treated group, rendering the intra-tumoral injection of rAF-IL12 to have very minimal/limited toxicity towards liver and kidney function.” It has to be rephrased as the enzyme levels were not lowered from a higher one in the same group of mice. It was an independent experiment where authors found lower enzyme levels.

·

Basic reporting

Najmuddin and co-workers studied the oncolytic effects of the recombinant NDV Strain AF2240-i expressing IL-12 (rAF-IL12) against HT29 colon cancer cells in vitro and in tumor-challenged NCr-Foxn1nu Nude mice. Their study proves that the recombinant NDV had better cytotoxicity effects than wild-type and could potentially be ideal for treatment of colon cancer in near future. The study comprises of an extensive set of experiments providing conclusive results. The literature and background content required to understand the scope of the study and its importance is clearly mentioned. They have also included evidences for choosing IL-12 as an immune-enhancing cytokine. In Line 106, the T cell differentiation information is not required as far as the study in concerned. It may require more experimental evidence to support that in the study and thus can be removed from the Introduction. The Materials and Methods section is very lucid and is understandable. The assays are clearly mentioned. The results and discussion have good coherence. The authors can try to shorten the discussion section a little bit.

Experimental design

Najmuddin and co-workers strategically designed the experiments which falls within the scope of the journal. The authors have pointed out the significance of generating the recombinant strain NDV as well as choosing IL-12 to be included in the construct. The authors can show the difference between the two constructs – wild-type and recombinant. It is missing in the introduction section. All assays performed are 72 hours post-infection and the authors did not mention any literature reference to back up. For Figure 1, the authors claim that viral copy number in HT29 is significantly higher in rAF-IL12 infected cells compared to wild-type AF2240-i. But their graph does not reflect any significant change. For Figure2, the graph does not show any significance value associated with it. For Figure 3, 4, and 5, the results section mentions Control while the figures talk about Negative Control. The labeling of samples used should be consistent throughout the article. For Figure 4, the quadrants are not explained well. They do not indicate what each quadrant is for. For Figure 7, the authors talk about the capability of rAF-IL12 to replicate inside tumor and organ tissues. The authors do not mention the reason behind the decision of choosing lung, spleen, kidney, and liver. Other vital immune organs can also have a strong influence which has not been reported in the study. Lines 327-332 also does not clearly mention what does dissemination show although their discussion section talks about it. Its better to be consistent with the flow of the results and observations. For Figure 8-11, the authors must re-write the result section. They need to clarify the outcomes of the observation. Currently, the result section just has the observation and no explanation on the findings. For Figure 12, the authors should explain their decision behind choosing the specific cytokines. Overall, all Figures in the images have faintly visible and missing error bars (Figure 7) and do not have the visually convincing error bars. The authors should put the images in high-resolution with visible error bars wherever probable.

Validity of the findings

The research findings are very convincing and promising. Authors have delved into each aspect of how a oncolytic viral particle can be tested. It will be great if the authors can explain why the recombinant variant has better viral copy number or has better response than the wild-type variant. It will be great if the authors can also perform a study showing the other possible variants of rAF-1L12 as IL12 plays a significant role in production of IL12 cytokine stimulating the immune response towards cancer.

---

## Round 0.2 · Minor Revisions

Dear Dr. Alitheen, one of our reviewers finds your rebuttal and revised manuscript satisfactory, however the reviewer still has few pertinent queries. Kindly address those and resubmit the manuscript.

Reviewer 1 ·

Basic reporting

This is a revised submission of the manuscript entitled “Oncolytic Effects of the Recombinant Newcastle Disease Virus, rAF-IL12, against Colon Cancer Cells In Vitro and in Tumor-challenged NCr-Foxn1nu Nude Mice”, concerning the anti-tumor effect of the recombinant newcastle disease virus, rAF-IL12. Authors have answered all the comments/concerns pointwise and improved the manuscript accordingly. The abstract, introduction and discussion sections are improved very well.

In the revised version of the manuscript, authors have added the use of recombinant NDV-IL12 strain for the oncolytic activity in previous studies. Same construct of the virus and IL-12 has been used for the inhibition of breast cancer. Another recombinant NDV strains have been published for colon cancer. The current study confirms these findings and is a good advancement from previous studies. However, few questions still remain unanswered.

Experimental design

Figure 5: I do understand author’s decision about performing the tumor treatment experiment at later time points in near future. However, is it possible for authors to check re-growth of the tumor after 28 days if the recombinant virus is not injected? Also, have authors tried any another route of injection if it makes any difference in anti-tumor activity?

Validity of the findings

Figure 1: This change in chart type does not answer the original question. On the basis of presentation, the previous graph was also fine. However, the question was regarding the calculation of infectious virus particles. Since qPCR gives signal with both infectious as well as defective virus particles, it can be only a supportive experiment, not a full proof. Authors should have the samples saved I assume. It should not take much time and effort to perform a plaque assay or TCID50 calculation to measure infectious virus particles.

Figure 12: Have authors also tested the gene expression levels of Fas, caspase-8, BID, BAX, and SMAD3 in non-cancerous cell lines, or normal mice? That will further prove authors finding that the recombinant virus does not cause any apoptosis in non-cancerous cell lines.

Additional comments

Line 505-509: It is not important to discuss the differences in different methods in discussion. These lines can be deleted.

I understand that in Conclusion section authors have mentioned all the experiments they wish to perform in near future. However, it does not seem appropriate to mention a list of experiments to be performed later. Specially, for very simple experiments such as plaque assays. It would be better if authors remove these future experiments from conclusion and mention either in results or discussion wherever the related experiment is discussed.

---

## Round 0.3 · accepted · Accept

Congratulations Dr. Alitheen, based on the revision of your manuscript and reviewer's recommendation, now your manuscript is accepted for publication.

Reviewer 1 ·

Basic reporting

no comment

Experimental design

no comment

Validity of the findings

no comment